A blockchain-based traceable and secure data-sharing scheme

Wang Zhenqi
Guan Shaopeng konexgsp@gmail.com
School of Information and Electronic Engineering, Shandong Technology and Business University , Yantai, Shandong , China
Ramachandran Sitharthan
Electronic publication date: 2023 Apr 4
Publication date: 2023
Volume: 9
Electronic Location ID: e1337
Received 2022 Sep 28; Accepted 2023 Mar 17
Copyright: © 2023 Wang and Guan
Copyright year: 2023
Copyright holder: Wang and Guan
License: This is an open access article distributed under the terms of the Creative Commons Attribution License, which permits unrestricted use, distribution, reproduction and adaptation in any medium and for any purpose provided that it is properly attributed. For attribution, the original author(s), title, publication source (PeerJ Computer Science) and either DOI or URL of the article must be cited.
License URL: https://creativecommons.org/licenses/by/4.0/

Keywords: Blockchain, Data sharing, Data tracking, Smart contracts, Inter-Planetary File System

Funding: The authors received no funding for this work.

==============================
The traditional data-sharing model relies on a centralized third-party platform, which presents challenges such as poor transaction transparency and unsecured data security. In this article, we propose a blockchain-based traceable and secure data-sharing scheme. Firstly, we designed an attribute encryption-based method to protect data and enable fine-grained shared access. Secondly, we developed a secure data storage scheme that combines on-chain and off-chain collaboration. The InterPlanetary File System (IPFS) is used to store encrypted data off-chain, and the hash value of encrypted data is stored on the blockchain. To improve data security, elliptic curve cryptography (ECC) encryption is performed before the hash value is stored. Finally, we designed a smart contract-based log tracking mechanism. The mechanism stores data sharing records on the blockchain and displays them in a visual form to meet the identity tracking needs of both data sharing parties. Experimental results show that our scheme can effectively secure data, track the identities of both parties sharing data in real-time, and ensure high data throughput.

Introduction

Data is a collection of symbols that record objective events in various forms such as words, numbers, and images. With the rapid development of information technology, data generated by all walks of life has exploded, becoming an important resource for human society. However, due to different business requirements, data is often segmented and stored by different departments and individuals, resulting in numerous independent datasets. This prevents data from flowing freely, creating “data silos” (Lu, Li & Xu, 2022), and resulting in low data utilization, hampering the value of data. Data sharing can make full use of data resources by sharing one’s data for others to use (Manzoor et al., 2021).

The traditional data-sharing model relies on a single third-party platform (Chen et al., 2019). The data owner uploads their data to the third-party platform, and the data demander pays the platform to obtain the data (Deepa et al., 2022). After the data-sharing is completed, the third-party platform pays the data owner a fee. This data-sharing model is easy to manage and maintain. However, data stored on a third-party platform is highly susceptible to tampering and resale, and the data owner will lose control of the data. Additionally, data-sharing records are the sole responsibility of third-party platform administrators and cannot provide data tracking capabilities for data owners, resulting in poor transaction transparency and lack of security (Wang, Tian & Zhu, 2018).

Blockchain (Wang & Li, 2021) is tamper-proof, open and transparent, and leaves traces throughout the process. It uses a chronological “chain” structure to store data, and any attempt to tamper with the data will cause the blockchain to break. Furthermore, the data stored on the blockchain is jointly maintained by all parties. Any operation information related to the data can be stored on the blockchain to realize the whole trace management of the data (Zhu et al., 2019). Blockchain provides new ideas for solving the problems existing in the traditional data-sharing model.

At present, blockchain-based data-sharing solutions have been implemented in various areas such as electronic medical records, smart grids, and supply chains (Xia et al., 2017b; Kumar, Marchang & Tripathi, 2022; Wan, Huang & Holtskog, 2020). However, the open and transparent nature of the data stored in the blockchain makes it difficult to guarantee the security of the data (Zhu et al., 2019). Moreover, the inability to delete data on the blockchain and the continuous increase in data volume significantly affects the efficiency of data query and storage (Li et al., 2022).

To summarize, existing data-sharing schemes utilize blockchain technology to ensure tamper-proofing, transparency, and traceability of shared data. However, the transparency feature of blockchain creates a risk of data leakage as shared data is disclosed to everyone on the chain. Additionally, the expanding scale of shared data results in increasing storage pressure on the chain. To address these challenges, we propose a blockchain-based traceable and secure data-sharing scheme that not only enables secure storage of shared data but also reduces the pressure of data storage on the chain. Our main contributions are summarized as follows: (1) We propose a data protection method based on attribute encryption. This method allows the data owner to customize the decryption authority of the encrypted data, enabling “one-to-many” encryption and decryption of shared data.

(2) We design a collaborative on-chain and off-chain data security storage scheme. The shared data is encrypted and stored off-chain in InterPlanetary File System (IPFS), and then the hash value of the shared data ciphertext is encrypted again and stored on the blockchain. This scheme not only protects data security but also relieves the pressure of data storage on the chain.

(3) We establish a visual data-sharing record system through smart contracts, which automatically stores data-sharing records on the blockchain and allows for identity tracking of both parties involved in the data-sharing process.

The rest of this article is organized as follows: “Related work” presents the related work, “Scheme design” outlines the proposed scheme, “Experiments and analysis” evaluates the scheme experimentally, and finally, “Conclusion” summarizes the work.

Related work

Currently, blockchain-based data-sharing solutions have become a prominent area of research in various fields. One of the early works in this area is the Medical Data Sharing System (MedRec), which aimed to facilitate patient access to medical records across hospitals using blockchain technology (Azaria et al., 2016). However, MedRec uses a Proof-of-Work (PoW) protocol for consensus, which incurs high computational costs and latency. Another healthcare blockchain project, MedicalChain (Albeyatti, 2018) aims to provide a transparent and patient-centric system for sharing electronic health records in a secure and auditable environment. Building on MedRec and MedicalChain, Singh et al. (2020) proposed a blockchain lightweight access control scheme for electronic medical records, enabling patients to upload and query their medical records through a designed smart contract API interface. In other fields, Majdalawieh et al. (2021) proposed a blockchain-based data-sharing scheme for supply chains to ensure data security sharing and traceability requirements, Liu, Sun & Song (2020) proposed a blockchain-based framework for sharing data in the food supply chain. Yang et al. (2022) proposed a blockchain-based data sharing framework for industrial IoT that records all data sharing activities in a blockchain for secure data sharing and auditing. Additionally, Chenli et al. (2022) designed a blockchain-based data-sharing platform that uses smart contracts to control access to data stored on the blockchain. While these schemes ensure the integrity and immutability of shared data using blockchain technology, the open and transparent nature of blockchain poses a risk of data leakage.

Encryption algorithms are a direct and effective way to secure data on the blockchain (Zhaoliang, Huang & Wang, 2021; Zhong et al., 2022). Dong et al. (2020) proposed using the AES (Advanced Encryption Standard) symmetric encryption algorithm to encrypt shared food information and store it on the blockchain to improve information security and credibility. However, using the same key for encryption and decryption in the symmetric encryption algorithm poses a key leakage risk. Zheng et al. (2018b) suggested using key keeper applications and blockchain for secure data sharing in the cloud. The blockchain verifies permissions, and key keeper applications store the symmetric key used for encryption. The data owner distributes the encryption key to key keeper applications, and data requesters obtain the key from key keeper applications to decrypt data. Baralla et al. (2021) proposed a blockchain-based supply chain data-sharing scheme that uses the RSA (Rivest-Shamir-Adleman) asymmetric encryption algorithm’s public key for data encryption and a private key for decryption, reducing the key leakage risk. However, the RSA asymmetric encryption algorithm has low efficiency, which affects system performance. Park et al. (2021) proposed the Medchain architecture, which uses proxy re-encryption to protect patient privacy. The patient’s private data is re-encrypted by a cloud-based proxy server node, which is vulnerable to attacks and poses a single point of failure risk. Zhang & Lin (2018) proposed the use of searchable encryption to secure shared data stored on the blockchain. However, most searchable encryption schemes are based on bilinear pairing and Diffie-Hellman difficulty assumptions, which means a public key encrypted ciphertext can only be decrypted by a corresponding private key (Guo, Yang & Yau, 2021). Zheng et al. (2018a) proposed a secure data-sharing scheme based on homomorphic encryption, which exploits the feature that homomorphic encryption can operate directly on the ciphertext to reduce the total encryption and decryption overhead of the system. However, the homomorphic encryption algorithm’s performance is slightly inferior to common encryption algorithms, which impacts the system’s performance.

In addition, as the scale of data on the blockchain increases, the inability to delete data on-chain leads to storage pressure on the blockchain. To address this issue, Yang et al. (2021) proposed a dual storage structure scheme that combines a centralized database and blockchain. This scheme reduces storage pressure on-chain by storing data off-chain in the centralized database, while the blockchain stores the hash values of the data. However, centralized databases are not secure, and data is vulnerable to tampering and reselling. To improve the efficiency of data-sharing, several studies have combined blockchain with cloud computing (Al Omar et al., 2019; Kaur et al., 2018; Xia et al., 2017a). Raw data is stored in the cloud to reduce the storage pressure on-chain. Nevertheless, most cloud-based data sharing systems are prone to the “single point of failure” problem, and third-party cloud service providers are semi-trusted, making secure data-sharing a challenge. To enhance the security of off-chain data, Guo et al. (2020) proposed a master-slave blockchain architecture to store data. In this architecture, shared data is stored in the slave chain, while the hash value of the data is stored in the main chain to prevent tampering. However, this architecture requires cross-chain collaboration in data storage and query, and its efficiency is low. IPFS is a decentralized storage system that uses multiple nodes to store multiple copies of data, and during data query, multiple nodes read the files simultaneously (Nizamuddin et al., 2019), greatly improving the efficiency of data query. To this end, Ye & Park (2021) proposed using IPFS to store vehicle data off-chain and storing the hash value of the data on the blockchain to improve query efficiency. However, unauthorized users can still retrieve data from IPFS by obtaining the hash of the data from the chain.

The above shows that existing data sharing schemes ensure data security on the blockchain through symmetric or asymmetric encryption techniques. However, symmetric encryption algorithms require the transmission of keys in a secure network, which has high system security requirements. Asymmetric encryption algorithms use different encryption and decryption keys. An encrypted message with a public key can only be decrypted by the corresponding private key, and cannot achieve one-time encryption that multiple participants can decrypt. The overall cost of the system is high.

Attribute-based encryption mechanisms are considered the most appropriate solution to the problem of secure data-sharing (Sun et al., 2020). It identifies the user with a set of attributes, and if the user’s attribute set matches the set required for ciphertext decryption, the user can decrypt the private key to recover the ciphertext to plaintext. This allows for “one-to-many” encryption and decryption of data and reduces the system overhead (Chen, Yin & Ning, 2022). However, in an attribute encryption system, multiple users with the same attribute set can exist in the system, which creates the possibility of unauthorized users decrypting data ciphertext.

To address the above issues, we propose using attribute-based encryption to protect shared data and achieve the “one-to-many” encryption and decryption requirements, where data encrypted by one user can be decrypted by multiple users, thus reducing the total encryption and decryption costs of the system. To reduce the storage pressure on the blockchain, the encrypted data is stored in IPFS and the hash value of the ciphertext is encrypted again using elliptic curve cryptography (ECC) encryption algorithm and stored on the blockchain to prevent unauthorized users from using the hash value on the chain to obtain the ciphertext and perform decryption operations.

Scheme design

System model

In this article, we propose a data-sharing scheme with the aim of improving data security and reducing storage pressure on the blockchain. The system architecture consists of four components as shown in Fig. 1.

Figure 1 System architecture.

IPFS: Provides off-chain storage services to data owners.

Authorization center: Generates the required parameters and keys for the data owner and the data demander.

Data owner: Performs attribute encryption operation on data, transmits the data ciphertext to IPFS, and obtains the hash value of the ciphertext.

Blockchain: Performs ECC encryption on the hash value of the off-chain data ciphertext and records the identity information of both parties involved in the data-sharing to generate a data-sharing log record.

Data demander: Obtain the encrypted hash value of the data ciphertext from the blockchain, download the data ciphertext from IPFS by decrypting the hash value, and finally restore the original data through attribute decryption.

Data protection method based on attribute encryption

Ensuring secure storage of data is a prerequisite for data-sharing. Symmetric encryption schemes are susceptible to key leakage, while asymmetric encryption schemes suffer from the problem of large encryption and decryption time overhead. Attribute encryption extends asymmetric encryption algorithms, which not only addresses the issue of key leakage in symmetric encryption schemes, but also allows the attribute private key generated for a user to meet the “one-to-many” encryption and decryption requirements. This means that the data encrypted by one user can be decrypted by multiple users, reducing the overall overhead of encryption and decryption. The formal definition of attribute encryption consists of the following four algorithms (Bethencourt, Sahai & Waters, 2007): (1) Setup (1λ,U)→(GP,MSK): The authorization center sets the security parameter λ and the global attribute set U, and generates the required public parameter GP and master key MSK through the initialization function Setup.

(2) KeyGen( Au,MSK) →SK: The user submits their attribute set Au, and the authorization center outputs the corresponding attribute private key SK for the user through the KeyGen key generation function and the master key MSK.

(3) Encrypt (M,GP,T)→CH: The data encryption user sets the access structure T and provides the public parameter GP and the data to be encrypted M. Then, the encryption function Encrypt is executed to generate the data ciphertext CH.

(4) Decrypt (SK,Au,CH)→M: The decryption user uses the attribute private key SK and attribute set Au as inputs to the decryption function Decrypt, which then restores the ciphertext CH to the original data M.

In the attribute encryption system, the access structure specifies which users corresponding to the attribute set can decrypt the data ciphertext. If a user’s attribute set satisfies the attribute set specified in the access structure, then the private key corresponding to this user’s attribute can successfully decrypt the ciphertext. Therefore, we utilize the attribute encryption method to secure the data under the chain, achieving the “one-to-many” encryption and decryption requirements. The flowchart for the off-chain data protection method based on attribute encryption is illustrated in Fig. 2.

Figure 2 Design of data protection method based on attribute encryption.

First, the authorization center generates the required parameters through initialization operations. Then, based on the attribute set, the authorization center generates the attribute private key for the data demander. The data owner customizes the access structure, specifying the set of attributes required for ciphertext decryption by the data demander, and encrypts the data. Next, the data owner uploads the encrypted data ciphertext to IPFS, obtains the hash value of the data ciphertext for ECC encryption, and records the identity information of both parties of the data-sharing to the blockchain to generate a data-sharing log record. Finally, the data demander downloads the data ciphertext stored in IPFS through the decrypted hash value and decrypts it using its own attribute private key.

On-chain and off-chain collaborative data security storage scheme

Although attribute encryption technology can fulfill the need for “one-to-many” data encryption and decryption, there is a risk of unauthorized data demanders being able to access the ciphertext of the data through the hash value, given that multiple data demanders with the same attributes can exist simultaneously in the system. To address this concern, we suggest encrypting the hash value of the ciphertext using the ECC encryption algorithm. This additional layer of encryption will safeguard the hash value of the ciphertext on the chain, thereby ensuring the security of the data.

ECC is a cryptographic system that relies on elliptic curves and the discrete logarithm problem. Compared to other encryption methods, ECC is known for its superior efficiency and can achieve a high level of encryption security with shorter keys. The elliptic curve E can be defined as (Koblitz, Menezes & Vanstone, 2000):

(1) E:y2+a1xy+a3y=x3+a2x2+a4x+a5

where ai(i=1,2,3,...,5)∈K and Δ≠0, K represents the defined rational number field and Δ is the discriminant of the elliptic curve equation:

(2) {Δ=−d22d8−8d43−27d62+9d2d6d2=a12+4a2d4=2a4+a1a3d6=a32+4a5d8=a12a5+4a2a5−a1a3a4+a2a32−a42

When Eq. (2) is satisfied by the elliptic curve E, Eq. (1) is referred to as the Weierstrass equation (Falcão et al., 2018). By further simplifying Eq. (2), we can derive the general expression for the elliptic curve:

(3) E:y2=x3+ax+b

where (a,b,x,y)∈Ep, Ep is a finite field and p is a large prime number.

ECC encryption involves selecting an elliptic curve Ep(a,b) and choosing a point G on the curve as the base point. Next, a private key r is selected, and the corresponding public key R=rG is calculated, where R and G are points on the elliptic curve Ep(a,b). To encrypt the plaintext, it is embedded into a point on the elliptic curve and the public key R is used to complete the encryption process. To decrypt the ciphertext, the private key r is utilized.

To protect data, it can be encrypted using the public key R, with higher efficiency for smaller amounts of data. As elliptic curves cannot be brute-force broken, ECC encryption provides a higher level of security at a lower computational cost. In light of the small data volume of the hash value and the high security requirement, we have opted to use the ECC encryption algorithm. In this study, we have utilized the key generation method from ECDSA (Elliptic Curve Digital Signature Algorithm) to produce the required public key R and private key r for ECC encryption. The specific process is shown in Algorithm 1.

Algorithm 1. Data security storage method for on-chain and off-chain collaboration.

1: Obtain the ciphertext hash value HCH of the data returned by IPFS;	
2: Generating elliptic curves Ep(a,b);	
3: Obtain the elliptic curve group (x,y);	
4: if data owner encrypts ciphertext hash value HCH then	
5:    The elliptic curve is obtained by Eq. (3), and y is obtained;	
6:    Get all the points that satisfy Ep(a,b), and get the base point G(x0,y0);	
7:    Generate the private key r for the data demander and compute the public key R=rG using the base point G(x0,y0);	
8:    The data demander transmits its own public key R to the data owner;	
9:    The data owner encrypts the data hash HCH using the public key R of the data demander;	
10:   Outputting the encrypted hash value ciphertext C and uploading it to the blockchain for storage;	
11:   The data demander decrypts the hash ciphertext C with his own private key r and obtains the decrypted hash value;	
12: end if	

Initially, the data owner obtains the hash value HCH of the data ciphertext in IPFS (line 1). Next, the system initializes the elliptic curve, generates the elliptic group, and completes the encryption algorithm initialization (lines 2–3). All points that satisfy the elliptic curve Ep(a,b) are calculated based on Eq. (3), and the base point G is obtained (lines 4–6). In the second step, the system generates the private key r and public key R for the data demander, and the data demander sends their public key R to the data owner (lines 7–8). Finally, the data owner encrypts the hash value HCH using the data demander’s public key R and stores the encrypted hash on the blockchain (lines 9–10). The data demander then decrypts the hash ciphertext on the blockchain using their private key r (line 11).

Smart contract-based log tracking mechanism

In a blockchain network, events are usually recorded in logs. As operational logs of the blockchain network are continually generated, displaying the logs visually can provide a more intuitive understanding of the entire process of events and enable identity tracking of both sides of data-sharing. Smart contracts (Wang et al., 2020), as an interface for interacting with the blockchain, have the characteristics of automatic execution and irrevocability, which facilitates the automatic storage and querying of data-sharing records. To this end, we have designed a smart contract-based log tracking mechanism that automatically stores data-sharing records on the blockchain through smart contracts and presents them in the form of visual logs. The smart contract’s interface design is presented in Table 1.

Table 1 Interface design of smart contracts.

Contract function	Contract logic	Contract method	Contract description	
User registration	Identity registration	userRegist()	The data owner or demander is registered in the system.	
Data write	Hash value sent	hashSent()	The data owner sends the encrypted hash value of the off-chain data to the data demander.	
	Hash value receive	hashReceived()	The data demander receives the encrypted hash value of the data owner’s off-chain data.	
Data query	Data owner query	querySent()	Data owners query data-sharing records on-chain.	
	Data demander query	queryReceived()	Data demanders query data-sharing records on-chain.	

Data owners and data demanders can upload and query data-sharing records by accessing the smart contract interface. The process of writing and querying data-sharing records is shown in Algorithm 2.

Algorithm 2. Algorithm for writing and querying data-sharing records.

1: if (userRegist(object)) then	
2:    The user registers successfully in the system and obtains an identity ID;	
3:    The data owner chains down the encrypted data and obtains the hash value HCH of the ciphertext CH;	
4:    Call Algorithm 1 to get hash ciphertext C;	
5: end if	
6: if (hashSent(OwnerID, C, DemanderID)) then	
7:    The data owner sends the encrypted hash value to the data demander;	
8:    if (hashReceived(DemanderID, C, OwnerID)) then	
9:       The data demander receives the encrypted hash value of the data owner;	
10:      Returns the transaction ID and block height blockNum of this data-sharing event;	
11:   end if	
12: end if	
13: if (querySent(transactionID)) then	
14:   The data owner queries the data-sharing record according to the transaction ID;	
15: end if	
16: if queryReceived(transactionID) then	
17:   The data demander queries the data-sharing record according to the transaction ID;	
18: end if	

First, the data owner and data demander register in the system (Lines 1–2). Then, the data owner uses Algorithm 1 to obtain the ciphertext of the encrypted data hash value (lines 3–5). When data-sharing is initiated between the data owner and the data demander, the blockchain records the identity information of both parties and the encrypted hash value, and returns the transaction ID of the data-sharing (lines 6–12). Finally, the data owner or data demander can query the data sharing record from the blockchain based on the transaction ID (lines 13–18).

Experiments and analysis

Experimental environment

In the experiments, we built the blockchain using Hyperledger Fabric (Kumar et al., 2021) and utilized IPFS as the storage platform under the chain. Table 2 shows the software configuration details of the experimental environment.

Table 2 Software configuration information of the experimental environment.

System component	Description	
Virtual machine version	VMware15 pro	
Hyperledger Fabric version	Hyperledger Fabric 2.4	
IPFS version	IPFS 0.14.9	
Smart contract language	Go 15.7	
Computer CPU	Intel Core i5-10300H @ 2.50 GHz	
Memory	16 GB	

We developed a user-friendly data-sharing and transmission system that connects with Hyperledger Fabric and IPFS through a web interface, based on the experimental environment configuration. This system allows data owners and data demanders to share data easily within the system.

Off-chain data security experiment

During the data sharing process, data owners store data off-chain in various formats, including video, audio, images, and text. Taking a text file as an example, the encryption result of off-chain data is shown in Fig. 3.

Figure 3 Encryption result of off-chain data.

Figure 3A shows the data encryption test interface, while Fig. 3B displays the encryption result of a text file. As shown, enabling attribute-based encryption data protection method presents the shared data in ciphertext, effectively preventing the leakage of shared data under the chain. Moreover, the shared data encrypted by one data owner can be decrypted by multiple data demanders, achieving “one-to-many” encryption and decryption of shared data and reducing the system’s overhead of repeatedly encrypting data. Subsequently, the encrypted data is uploaded, and Fig. 4 shows the hash value of the encrypted data returned by IPFS.

Figure 4 Comparison of the hash value of ciphertext before and after encryption.

Figure 4A displays the result of the unencrypted hash of the data ciphertext, while Fig. 4B shows the result of the ECC encryption of the hash of the data ciphertext. In Fig. 4A, when the hash value is not encrypted, unauthorized data demanders can download the data ciphertext stored in IPFS directly via the hash value. If the set of attributes of this data demander matches the set of attributes required for the ciphertext decryption, they can decrypt the data ciphertext. In contrast, Fig. 4B shows the encrypted hash value of the data ciphertext stored in IPFS in the form of ciphertext. The ECC encryption algorithm uses a public key for encryption and a private key for decryption. Only the data demander with the decryption private key can decrypt the hash value and obtain the ciphertext of the data in IPFS.

To track the identity of both parties involved in the shared data, we utilize smart contracts to automatically record and track the data-sharing process, and present it in the form of visual logs. Figure 5 shows an example of the visual log.

Figure 5 Shared records on the blockchain.

Figure 5 illustrates the visual log of each sharing record between the data owner and data demander. This log includes details such as the block number, the identity of the sharing parties, and the time of the sharing. This provides real-time tracking of the identities involved in the data sharing process. Additionally, to reduce the storage pressure on the chain, our scheme offloads the storage of shared data to off-chain platforms. Table 3 presents the storage space occupied by the shared data on the chain.

Table 3 Data space occupancy table.

Size of raw data	No off-chain storage solution	Proposed scheme	
1 MB	1 MB	0.1 MB	
10 MB	10 MB	0.1 MB	
20 MB	20 MB	0.1 MB	
30 MB	30 MB	0.1 MB	
40 MB	40 MB	0.1 MB	

As presented in Table 3, the proposed on-chain and off-chain collaborative data security storage scheme can significantly decrease the occupancy rate of on-chain storage. Furthermore, the proposed secure data storage solution occupies the same amount of space on the chain as the volume of raw data increases. This is because only the encrypted hash of the data is stored on the chain, and the hash size remains constant regardless of the amount of data stored off-chain (Kumar et al., 2021).

Performance analysis of the system

During the process of data-sharing, the system incurs performance overhead which mainly consists of the execution cost of data encryption and decryption, and the transaction cost of the chain. We measure these costs, including the encryption and decryption overhead of raw data, as well as the encryption and decryption overhead of hash values. For instance, using the data in Table 3 as an example, we measure the encryption and decryption overhead of the raw data, while the overhead of hash values encryption and decryption is explained later. To ensure experimental data reliability, the experiment was executed 100 times and averaged, as depicted in Figs. 6 and 7 for encryption and decryption overhead of original data, respectively. Additionally, to test the performance of the proposed scheme during encryption and decryption of original data, it was compared to the schemes proposed by Dong et al. (2020) and Park et al. (2021). Dong et al. (2020) proposed the use of AES symmetric encryption algorithm, while Park et al. (2021) suggested the use of proxy re-encryption technique for data privacy protection. These two schemes were proposed recently and are similar to the technical architecture of the proposed option.

Figure 6 Encryption overhead of raw data.

Figure 7 Decryption overhead of raw data.

In Figs. 6 and 7, it is evident that AES encryption and decryption incur the lowest execution cost, while proxy re-encryption incurs the highest execution cost. This is mainly because the AES encryption and decryption process uses a byte round-robin conversion operation, making the encryption and decryption of original data relatively fast. However, since AES encryption and decryption utilize the same key, data owners need to share the key with data requesters in advance, which can easily lead to key leakage. On the other hand, the proposed scheme does not require sharing the key during original data encryption and decryption, making it more secure. Furthermore, the proposed scheme is more efficient in original data encryption and decryption compared to the proxy re-encryption technique. This is because the proxy re-encryption technique needs to convert the ciphertext to different ciphertexts through a proxy server for different data requesters to decrypt, which increases the implementation cost of the system.

Regarding the encryption and decryption overhead of hash values, we also measures it. The experiment is executed 100 times to obtain the average value, and the results are presented in Table 4.

Table 4 Encryption and decryption overhead of hash values.

Type	Proposed scheme	
Encryption time	156.24 μs	
Decryption time	86.54 μs	

Based on Table 4, it can be observed that hash value encryption and decryption have a minimal impact on the system’s efficiency. The designed ECC encryption algorithm not only effectively protects the chain hash values from leakage but also encrypts and decrypts the hash values without significantly reducing the system’s efficiency.

Data throughput refers to the number of transactions completed per second on the blockchain and reflects the transaction cost overhead on the chain. In this study, we measured the throughput of shared data storage and query in the blockchain network and compared it with the scheme without under-chain storage and the Guo et al. (2020) scheme. Guo et al. (2020) proposes the use of a master-slave blockchain architecture to store data. Among them, the slave chain stores the shared data, and the master chain stores the hash of the data, which is similar to the architecture of the proposed scheme. To ensure the reliability of the experimental data, the experiment was repeated 100 times to obtain the average value, and the results are presented in Fig. 8.

Figure 8 Data throughput experimental results.

Figure 8 demonstrates that the throughput of the scheme without under-chain storage gradually decreases as the data size increases, and the throughput in the Guo et al. (2020) scheme is even lower. Meanwhile, the throughput of the proposed scheme remains constant and is higher than that of the chainless under-storage scheme. This is because the Guo et al. (2020) scheme requires cross-chain operations between the master and slave chains during data storage and query, which unavoidably decreases the efficiency of blockchain storage and query. In contrast, the proposed scheme only stores the encrypted hash value, which is significantly smaller than the shared data directly on the chain. As a result, the proposed scheme has higher throughput.

Data security comparison

We have compared our scheme with other related schemes with regards to off-chain data security, on-chain data security, data integrity, data tracking, trustfulness, authorization, authentication, reliability, and validation. The results are presented in Table 5.

Table 5 Comparison results of similar schemes.

Features	Our scheme	Zhang’s scheme	Guo’s scheme	Ye’s scheme	Yang’s scheme	
Data integrity	Yes	Yes	Yes	Yes	Yes	
On-chain data security	Yes	Yes	No	No	No	
Off-chain data security	Yes	No	No	No	No	
Data tracking	Yes	No	No	No	No	
Trustfulness	Yes	Yes	Yes	Yes	No	
Authorization	Yes	Yes	Yes	No	No	
Authentication	Yes	No	Yes	No	No	
Reliability	Yes	Yes	No	Yes	No	
Validation	Yes	Yes	Yes	No	No	

(1) Off-chain data security

Our data security protection method is based on attribute encryption, in which the data owner encrypts the original data by customizing the access structure to achieve fine-grained access control to the data off-chain, thereby enhancing data security. (2) On-chain data security

We use the ECC encryption algorithm to store the data hash as a ciphertext on the blockchain. The data ciphertext can only be downloaded from IPFS if it is decrypted, preventing unauthorized access to the storage address of the data off-chain and enhancing on-chain data security. (3) Data integrity

Our proposed collaborative data storage method encrypts and stores off-chain data in IPFS, which can be stored permanently. Moreover, the encrypted hash value stored on the chain cannot be tampered with, ensuring data integrity. (4) Data tracking

Through the smart contract interface, data-sharing records are tamper-proof stored on the chain, allowing real-time tracking of the identity of both parties sharing data. (5) Trustfulness

Our data security storage scheme stores shared encrypted data by IPFS, while the hashes of the encrypted data are stored by the blockchain. Based on the uniqueness of the hash value, data demanders can compare the hash value of the data under the chain with the decrypted hash value on the chain to verify whether the data is trustworthy, enhancing trustfulness. (6) Authorization

Our scheme ensures that only authorized data demanders can decrypt the cryptographic hashes stored on the blockchain and download the data ciphertext stored in IPFS. Additionally, the data ciphertext can be decrypted only if the set of attributes of the data requester matches the set of attributes required for the decryption of the ciphertext, ensuring data confidentiality. (7) Authentication

In our scheme, both data owners and demanders must be authenticated by the system to access the system. After authentication, the system provides keys. Our designed data tracking mechanism fully records shared data on the blockchain for easy accountability, ensuring system security. (8) Reliability

In this article, we store the original data off-chain, and the uniqueness and irreversibility of the hash value ensures the reliability of the off-chain data, which is further guaranteed by the on-chain storage of cryptographic hash. The cryptographic hash of the data on the chain cannot be changed, ensuring the reliability of the data. (9) Validation

In this article, the data stored on and off the chain is presented in ciphertext format. Unauthorized or unverified data demanders querying the blockchain for data will only get the hash ciphertext and will not be able to download the original data ciphertext stored in IPFS. Therefore, the original data ciphertext can only be decrypted by authorized parties, achieving on-chain and off-chain data verifiability.

It is evident that our solution not only ensures secure storage and sharing of data, but also enables tracking the identity of the data sharer.

Conclusion

In this article, we propose a traceable and secure data-sharing scheme based on blockchain technology. Our solution features a data protection method based on attribute encryption to enable fine-grained access control for shared data. To enhance data security, we employ a collaborative on-chain and off-chain data storage scheme, which also alleviates storage pressure on the chain. Additionally, we design a log mechanism based on smart contracts to facilitate identity tracking for both data sharers. Our proposed scheme is evaluated for performance and security, demonstrating higher data throughput and security, as well as lower data encryption and decryption overhead costs. Furthermore, it enables real-time identity tracking for both parties involved in data sharing, while ensuring data encryption protection both on and off the chain.

Supplemental Information

Supplemental Information 1 Experimental code.

Click here for additional data file.

Additional Information and Declarations

Competing Interests

Author Contributions

Data Availability

The authors declare that they have no competing interests.

Zhenqi Wang performed the experiments, analyzed the data, performed the computation work, prepared figures and/or tables, and approved the final draft.

Shaopeng Guan conceived and designed the experiments, analyzed the data, prepared figures and/or tables, authored or reviewed drafts of the article, and approved the final draft.

The following information was supplied regarding data availability:

The code is available in the Supplemental File.

The raw data is available at Zenodo: Zhenqi Wang, Shaopeng Guan. (2023). Raw data. https://doi.org/10.5281/zenodo.7529808.

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
