# Peer review of "A blockchain-based traceable and secure data-sharing scheme"

_PeerJ Computer Science, doi:10.7717/peerj-cs.1337_

## Round 0.1 · original submission · Major Revisions

Several typo errors can be found all through the manuscript.
Precise graphs and data must be provided.
Comparative analysis must be provided.

Reviewer 1 ·

Basic reporting

In this paper, the authors designed an attribute encryption-based data protection method to achieve fine-grained shared access to data. As a part of this method, they designed a secure data storage scheme with on-chain and off-chain collaboration. While the encrypted data off-chain is stored in the Inter Planetary File System (IPFS), the hash value of the encrypted data is stored on the blockchain.

- In my opinion, the related work section seems to me to be poor, so it would need to be extended. In the literature, there are many blockchain-based secure data-sharing schemes.

Experimental design

In addition to the data security comparison, a cost consumption comparison of the existing and proposed systems using metrics such as execution cost and transaction cost is required.

- Instead of just the scheme names, Table 4 should include the reference for other schemes.

- The authors just handled data integrity, on-chain data security, off-chain data security, and data tracking parameters for the security comparison. But according to my opinion, I advise using trustfulness, authorization, authentication, reliability, validation, and decentralized parameters for deep comparison.

Validity of the findings

- Actually, the structure of the proposed scheme is valuable, but it needs to prove the benefit and superiority of the scheme more clearly by using additional comparison tables.

Additional comments

-

Reviewer 2 ·

Basic reporting

- It is not clearly explained how the data demander can decrypt the encrypted hash value. To be able to decrypt the encrypted hash value, the data demander needs to have the corresponding private key. If the authors believe that there is no need for private key, then anyone can decrypt the hash value.

- It is not clearly explained which encryption scheme based on ECC is used. Plain elliptic curve is not used alone for encryption. The encryption method described in the manuscript (Algorithm 1 step 7) is not correct. I suggest the authors to present a complete ECC based encryption scheme.

- Line 91: "However, homomorphic encryption is only suitable for numerical data that needs to be calculated."
This statement is not correct. All data can be represented as a binary data which can be received as an input to the encryption algorithm. Same thing is valid for all encryption schemes not only for homomorphic encryption.

- There are some technical typos:
Line 124: "Authorization center: Initializes encryption algorithms". Actually, authorization center generates parameters and secret keys. Doesn't initialize encryption algorithms
Line 126: "Data owner: perform attribute encryption operation on data, transmit data ciphertext to IPFS, and obtain the hash value of ciphertext." This explanation doesn't seem to be aligned with Figure 1.

- Line 135: "The existing symmetric encryption 136 scheme is not highly secure,"
This argument is not a correct argument.

- Line 136: "while the asymmetric encryption scheme suffers from the problem of large encryption and decryption time overhead."
Same issue is also valid for attribute-based encryption.

- Line 146. "(2) KeyGen(Au,146 MSK) -> SK: The user submits his attribute set Au and the master key MSK, and outputs the corresponding attribute private key GP, through the key generation function KeyGen."
MK is not known by the user. The output is the SK not the GP.

- Line 151: "(4) Decrypt(SK,Au,151 CH) -> M: The decrypted user inputs attribute private key SK and attribute set Au, and restores the ciphertext CH to the original data M through the decryption function Decrypt."
"decrypted user" is not a correct term.

- Line 160: "160 Then, according to the attribute set, the data demander generates the attribute private key"
The authority generates the secret key, not the data demander.

- Line 182: "When data needs to be protected, it can be encrypted by mapping it onto an elliptic curve E."
The curve is not used alone for encryption. There are some encryption schemes based on EC. This encryption definition is not correct.

Experimental design

no comment

Validity of the findings

- According to the proposed solution, attribute-based encryption is used to allow to share the data with only some data demanders having some special attributes. Also, ECC-based encryption is used to share the hash of the data with only specific data demander(s). Considering both these main steps, the value of usage of attribute-based encryption algorithm is not clear. I suggest the authors make this point clearer.

- Usage of ECC-based encryption is not complete in the manuscript. How the data demander can have the decryption (private keys) should be explained. Which ECC algorithm is used needs to be explained.

Reviewer 3 ·

Basic reporting

no comments

Experimental design

1.A suggestion to throw some light on 'Data Sharing Transmission System' in experimental environment.

2. Authors can prefer the word ' proposed scheme/ proposed method to 'our scheme'

Validity of the findings

The authors has concluded:
[Conclusion Section]
"At the same time, it also has higher data throughput and security and can
perform real-time identity tracking for both parties of data sharing."

Authors can add some findings as justification for the statement mentioned in conclusion .

Figure 6 can be presented more precisely to justify the statement.

The authors can improve the results by providing run time analysis of proposed and compared schemes.

---

## Round 0.2 · Minor Revisions

Revise the manuscript as per reviwer comments.

Reviewer 2 ·

Basic reporting

no comment

Experimental design

no comment

Validity of the findings

I have a concern about the complexity of Algorithm 1. In Step 4, the condition in the "while" loop is "x \le p-1". When we consider the length of the prime number (p) is 256-bit, this means this loop needs to be executed nearly 2^256 times which is impossible in practice. It would be easier just to refer to an existing ECC encryption scheme instead of trying to provide the details, or the "while" loop should be reconsidered. The vital information that needs to be provided in the paper is that the data demander generates the public/private key pair and sends the public key to the data owner, which has been mentioned in the revised version.

Additional comments

Thanks for the revised version that addresses almost all my comments.

---

## Round 0.3 · Minor Revisions

Your submission is nearly ready for publication. However, you need to clarify your contribution to the field, and perform a thorough proofread.

---

## Round 0.4 · Minor Revisions

The reviewers' comments have been addressed. However, there are 2 remaining issues:

1) Please add some further discussion on your work in the context of current literature.

2) A thorough proofread is required.

Please address these and resubmit. Thank you.

---

## Round 0.5 · accepted · Accept

There are just a couple of typos, but all other issues have been addressed so this is ready for publication in my view.